# RelEval: A Structured Benchmark for Logical and Relational Reasoning in LLMs

## Abstract

We introduce RelEval, a benchmark for evaluating large language models (LLMs) in logical reasoning over complex relational structures. Such reasoning underpins applications where LLMs generate or query structured graphs, including network infrastructure, knowledge bases, and business process schemas. Our framework enables fine-grained control of task difficulty by varying the number of objects, relations, and the depth of relational chains. RelEval encompasses three complementary tasks: (1) *Plan Generation*, requiring construction of valid directed relational graphs under structural constraints; (2) *Consistency Detection*, detecting inconsistencies in relational structures; and (3) *Comparison Question*, assessing the validity of queried relationships. We also test models' self-correction by prompting them to verify and refine their answers. We evaluate DeepSeek R1, Gemini 2.0 Pro, Gemini 2 Flash Thinking, GPT-4.5, GPT-4o, Llama 3.1 405B, O3-mini, O1, and Claude 3.7 Sonnet, finding large performance gaps linked to model scale and architecture. While recent reasoning-focused models excel on simpler cases, they struggle with more complex configurations requiring deeper reasoning.

## 1 Introduction

Recent advances in large language models (LLMs) have sparked interest in their ability to perform complex reasoning tasks (Guo et al., 2025; OpenAI, 2025; Jaech et al., 2024). While current work showcases increased reasoning skills in math and coding skills, there are several real-world problems which require reasoning skills that are more algorithmic in nature and often computationally expensive. We still lack an in-depth understanding of how current models perform and how their performance scales with the size of these problems. Logical relational reasoning falls into in this category, which requires models to plan or validate relationships between a large number of objects. For example, in supply chain operations research and business planning (Tsouros et al., 2023; Li et al., 2023), there exist several relational planning problems in which each object is a task or an event, and a relationship is a constraint that specifies whether a task should occur before or after another task. Similar examples also exist in code analysis (Neamtiu et al., 2005), and agentic systems (Wu et al., 2023; Fourney et al., 2024) where dependencies are introduced and task planning and orchestration are required. LLM-based chatbots and agentic AI systems has transformed how people interact with natural language. This evolution necessitates more rigorous evaluation of relational reasoning to better understand the advantages and limitations of LLMs.

In this work, we introduce RelEval, the first benchmark specifically designed to assess both the generation of structured relational graphs and the subsequent reasoning about them. Our framework fills an important gap by focusing on the entire pipeline, from the autonomous construction of relational graphs to the critical examination of their logical consistency, which is central to many real-world applications.

The structured form of RelEval enables dynamically varying parameters such as the number of objects, comparative relations, and the minimum depth of relational chains, in either symbolic relation format or natural language converted descriptions. This facilitates a fine-grained analysis of model performance across a wide spectrum of difficulties. This approach mirrors the challenges faced by intelligent agents operating in unpredictable environments, where the ability to generate and validate continuously updated relations is critical for successful deployment. Dynamic data generation also

offers the advantage that it allows for generating fresh or private copies of the benchmark with controlled difficulty, in cases where benchmark memorization becomes a concern.

RELEVAL consists of three complementary tasks: (1) *Plan Generation* - Models must construct valid directed graphs with comparative relations meeting specified structural constraints. (2) *Consistency Detection* - Does a given graph contain cycles or contradictions? What are they?. and (3) *Comparison Question* - Is a relation valid for a given graph?. Through a set of RELEVAL experiments, we present a comprehensive evaluation of nine state-of-the-art models, encompassing both instruction-tuned and reasoning-capable LLMs. Our analysis uncovers performance disparities that correlate with model scale and architecture, providing valuable insights into the evolving capabilities of LLMs.

The evaluation reveals a significant gap between the reasoning-based and instruction-based models in the Plan Generation task. This difference stems from the reasoning models' ability to identify a straightforward generation technique where objects maintain logical relationships, such as $A > B > \cdots > Z$. For the Consistency Detection and Comparison Question tasks, the gap persists, primarily between O1 and O3-mini and the other models. In particular, Consistency Detection remains the most difficult task, where even state-of-the-art reasoning models face challenges as the size of the problem increases. Specifically, the performance drop for this task happens much earlier for models like DeepSeek R1 and Gemini 2 Flash Thinking.

Overall, RELEVAL serves as both a diagnostic tool and a catalyst for future research, pushing the boundaries of what is achievable in logical relational reasoning with LLMs. We will open source the benchmark, the code for data generation and model evaluation to enable future research on LLM evaluation and reasoning upon publication.

## 2 RELATED WORK

**Logical Reasoning in LLMs.** Logical reasoning has been studied as a fundamental capability of large language models in the context of inductive reasoning, propositional and first-order logic (Patel et al., 2024; Ryu et al., 2025; Parmar et al., 2024; Xu et al., 2024), and a diverse set of logical puzzles (Lin et al., 2025; Shah et al., 2024). Similarly to RELEVAL, these works often involve one or multi-hop reasoning as a way of testing whether models can retrieve and reason upon more than one statement or fact presented in context (Trivedi et al., 2022; Yang et al., 2018; Patel et al., 2024; Schnitzler et al., 2024; Zhu et al., 2024). For example, HotpotQA includes a set of linguistic facts in each example, and then asks related factoid comparison questions. However, recent work has also shown that the accuracy of models in conducting multi-hop reasoning drops as the length of context increases (Levy et al., 2024; Balachandran et al., 2024) or with the number of inference rules (Patel et al., 2024).

Relational reasoning in LLMs has not been studied extensively yet since until recently the task was beyond the capabilities of the state-of-the-art, especially for larger problem sizes. Preliminary work (Li et al., 2024; Alotaibi et al., 2024) with a few-hop queries has shown that larger models can conduct simple logical queries on graphs to extract relationships between objects or their attributes. Advancements in test-time compute and post-training (or self-training) via Reinforcement Learning (RL) (Lightman et al., 2023; Wang et al., 2024c; Gulcehre et al., 2023) have shown to significantly improve general reasoning skills for problems like math and coding (Guo et al., 2025; OpenAI, 2025; Jaech et al., 2024). Qualitatively, these methods lengthen the generations of models in the form of extended reasoning traces, which contain notions of self-reflection, backtracking, and exploration. In this work, we include several of the state-of-the-art reasoning models, to better understand whether these techniques generalize to more complex forms of reasoning, such as inference and cycle detection in larger graphs. More relatedly to our work, Abbe et al. (2024) has shown that intermediate steps in detailed scratchpads in transformers can help address barriers introduced by high connectivity and longer chains in difficult queries.

**Planning in LLMs.** Our work can also be situated in the context of studying planning abilities of LLMs, which has been evaluated extensively in previous work (Valmeekam et al., 2024; 2023; Wang et al., 2024b; Zheng et al., 2024) with benchmarks like PlanBench (Valmeekam et al., 2024), NaturalPlan (Zheng et al., 2024), CogEval (Momennejad et al., 2023), APR (Lin et al., 2024). CogEval in particular also looks at routing problems in graphs, but does not extend to cycle detection or consistency checks of relationships. (Lin et al., 2024) uses DAGs to optimize execution time

and parallel vs sequential execution, but only focuses on numerical schedule outputs (minimizing time cost).Wang et al. (2024a) instead formulates a task in a multimodal setting that asks multiple-choice questions about the relationships between pairs of objects, which can be inferred from other relationships already presented in context.

Most importantly, given the high computational complexity of the problems in RELEVAL, our benchmark also contributes to an emerging line of work that studies how transformers solve NP-hard and NP-complete problems (Fan et al., 2023). With the increased generality of current architectures, it is important to understand in detail the limitations of transformers in solving such complex problems and how they scale with the difficulty of the problem.

## 3 RELEVAL CONSTRUCTION AND DATA GENERATION

Our approach focuses on developing a dynamic benchmark to probe the logical and relational reasoning capabilities of LLMs. RELEVAL is designed to ensure broad coverage of reasoning tasks while maintaining control over the complexity of relational structures. We adopt a strategy that synthesizes relational graphs with controllable complexity. By varying parameters like object count, relation density, and chain depth, we create diverse, scalable tasks to systematically evaluate reasoning.

### 3.1 PLAN GENERATION

The Plan Generation task is motivated by real-world applications where models must autonomously construct complex relational structures—such as network topologies or business process models—without human intervention. The primary goal is to evaluate the models' ability to produce consistent, accurate, and complete plans that adhere to specified constraints.

*Task Design*: In this task, each model generates a relational graph based on given specifications (i.e., number of objects and relations). The evaluation covers three dimensions: the overall accuracy in generating the required relations, logical consistency of the constructed graph, and the absence of redundant relations. This evaluation framework ensures that the models are not only capable of constructing a plan but also of maintaining structural integrity and coherence as the complexity of the task increases.

*Data Generation*: We generated datasets of varying complexity using a Python program that systematically pairs the number of object with the number of relations. The dataset comprises pairs with object counts ranging from 3 to 50. For each object configuration, we varied the number of relations from minimal (equal to the object count $n$) to dense (approaching $n(n-1)/2$, with practical upper limits), creating a diverse testbed for evaluating language model performance on graph-structured generation tasks.

### 3.2 CONSISTENCY DETECTION

In many applications, especially those involving critical infrastructure or knowledge management, identifying inconsistencies such as cycles or contradictory relations is important. The Consistency Detection task is designed to assess a model's ability to scrutinize a given graph and pinpoint logical errors.

*Task Design*: In this task, models are presented with relational graphs that may contain cycles or contradictions. The evaluation focuses on two main aspects: the model's ability to detect the presence of inconsistencies, and to correctly identify and extract these inconsistencies. This involves assessing the effectiveness in detecting true inconsistencies while minimizing false alarms, and providing a balanced measure of its detection capabilities. By varying the number of objects, relations, and the minimum cycle length (how many steps are needed for the cycle to close), we challenge the models with increasingly complex scenarios, ensuring a comprehensive assessment of their verification capabilities.

*Data Generation*: We use the Python NetworkX library to construct relational graphs, including both positive (acyclic, consistent) and negative (cyclic, inconsistent) cases. Graphs are generated with controlled complexity, varying the number of objects and the density of relations while ensuring

complete connectivity. For cyclic cases, we enforce minimum cycle lengths in the graph to assess multi-step reasoning abilities. These graphs are converted into randomized textual statements where: (1) object labels are arbitrary and unrelated to their structural positions, (2) relations use mixed notation (randomly choosing between $>$ or $<$), and (3) statement order is shuffled. This design prevents shortcut pattern matching, and requires logical reasoning to reconstruct the complete relation network and identify cycles regardless of their surface representation. An example base graph is shown in Figure 5 in Appendix.

### 3.3 COMPARISON QUESTION

The Comparison Question task simulates scenarios where models must interpret a pre-existing relational graph and make informed judgments about specific relations. This task is pivotal for applications such as automated reasoning in databases or dynamic knowledge base querying.

*Task Design*: In this task, models are required to evaluate relational statements and categorize them as *True*, *False*, or *Unknown*. The *Unknown* category is particularly significant as it assesses a model's ability to handle incomplete information, i.e., situations where the given graph is insufficient to answer the query about the specified relation. We challenge the models with increasingly complex problems by examining performance based on the number of objects, relations, and the required inference depth. This design tests not only direct reasoning capabilities but also the model's decision-making process in complex and edge cases.

*Data Generation*: We use the same graph generation algorithm as the Consistency Detection task. For each graph, object pairs are randomly selected at specific path distances and labeled as *True* if the comparison matches the graph, *False* if it contradicts it, or *Unknown* if no path exists between them.

### 3.4 SELF CORRECTION

We also evaluate models' self-correction, or their ability to assess and revise outputs. This reveals how internal confidence shapes decisions and is essential for developing models that reason reliably and adapt under uncertainty.

*Task Design*: Following the completion of the Consistency Detection and Comparison Question tasks, models are given a follow-up prompt: *Are you sure?*. We assess two key aspects: the degree to which initial decisions are revised, and the impact of these revisions on overall accuracy. Specifically, we examine whether self-correction leads to improved outcomes or introduces additional errors.

### 3.5 NATURAL LANGUAGE FRAMING

In real-world scenarios, relational information is often conveyed in unstructured or natural language form. To better evaluate model performance in such settings, we assess their ability to interpret and reason over inputs expressed in natural language, rather than in mathematical form.

*Task Design*: We recast the Consistency Detection and Comparison Question tasks from their original symbolic format ($>$ and $<$) into natural language descriptions grounded in real-world scenarios. The prompts are rewritten to establish the scenario-specific context. We test two scenarios: detecting circular dependencies in software packages and identifying dependency risks in financial institutions.

## 4 EVALUATION

In this section, we evaluate RELEVAL across a range of state-of-the-art models. Unless noted otherwise, all experiments use five independent runs, reporting average performance to account for non-determinism and improve robustness. Models are tested under their respective settings, with detailed configurations in Table 2 (Appendix).

*Instruction-based models*: Llama 3.1 405B, Gemini 2.0 Pro, GPT-4o, GPT-4.5.

*Reasoning models*: O1, O3-mini, DeepSeek R1, Gemini 2 Flash Thinking, Claude 3.7 Sonnet.

We employ task-specific metrics to rigorously assess performance under varying computational demands:

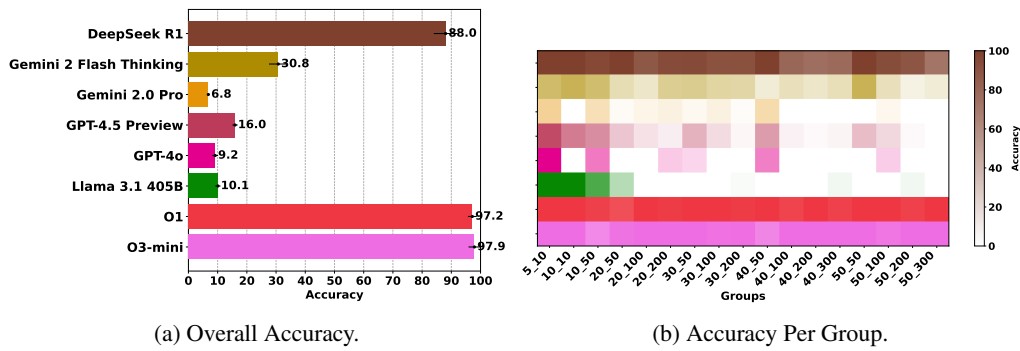

(a) Overall Accuracy.                           (b) Accuracy Per Group.

Figure 1: Overview Accuracy of different models for the Plan Generation task.

**Plan Generation.** For the Plan Generation task, we use *Accuracy*—the proportion of correctly generated plans. A perfect score (100%) means the model produces the required number of relations and objects without inconsistencies and duplications. *Accuracy* is further broken down into:

- *Consistency*: whether generated relations are logically consistent (100% = no inconsistencies).
- *Duplicates*: absence of repeated relations (100% = no duplicates).
- *Object Numbers* and *Relation Numbers*: match between generated counts and task specifications (100% = full compliance).

**Consistency Detection.** For the Consistency Detection task, we use the *F1 score*, derived from *Precision* and *Recall*, to assess decision correctness. We evaluate: (1) whether the model correctly detects the presence or absence of inconsistencies, and (2) when present, whether it identifies all of them. Specifically:

- *Precision*: for consistency cases, whether the model correctly reports consistency; for inconsistency cases, the proportion of reported inconsistencies that are correct.
- *Recall*: for consistency cases, always 1 (as there is no inconsistency to find in the ground truth); for inconsistency cases, the proportion of ground-truth inconsistencies correctly identified.

**Comparison Question.** For the Comparison Question task, we use *Accuracy* to evaluate whether the model correctly determines the truth of given relations:

- *True*: the comparison statement is entailed by the graph.
- *False*: the graph entails the comparison in the opposite direction.
- *Unknown*: the graph provides insufficient information to determine the relation, capturing the model's ability to handle ambiguity or incompleteness.

**Self Correction.** For the Consistency Detection and Comparison Question tasks, we evaluate self-correction by adding a second-turn prompt—*Are you sure?*—and measuring changes in the model's output. The output format and evaluation metrics remain consistent in the second turn.

**Natural Language Framing.** Finally, we recast Consistency Detection and Comparison Question inputs from symbolic ($>$ and $<$) to natural language descriptions. To ensure comparability, we retain the same output format and metrics.

### 4.1 PLAN GENERATION RESULTS

We begin with the Plan Generation task as a warm up task. In this task, each model generates a consistent list of relations based on the number of objects and relations specified in the prompt, and is evaluated over three independent runs. Our dataset comprises 142 examples, with objects ranging from 3 to 50 and relations ranging from 3 to 300. For each group of object numbers, the number of relations is incrementally increased from a minimum edge with a minimal step to a maximum reach evenly. Detailed configurations of these 142 examples and prompt template are provided in Appendix.

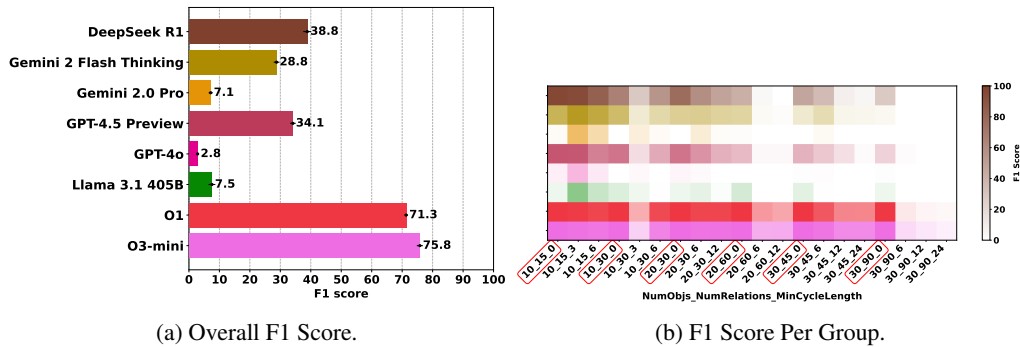

(a) Overall F1 Score.                    (b) F1 Score Per Group.

Figure 2: Overview F1 Score of different models for the Consistency Detection task.

**Overall Results.** We first present the overall accuracy (averaged across all runs) for each model, as shown in Figure 1a. The figure highlights a substantial performance gap between reasoning models and instruction-based ones. For instance, the O3-mini and O1 models achieve notable accuracies of 97.9% and 97.2%, respectively, while DeepSeek R1 attains 88.0%. Interestingly, the Gemini 2 Flash Thinking model, despite being a reasoning model, only achieves 30.8%, which is significantly lower. However, it still performs approximately twice as well as the best instruction model GPT-4.5.

Next, we present a more fine-grained analysis of accuracy by dividing the settings into bins and reporting each model's performance, as illustrated in Figure 1b. In this figure, the x-axis is labeled as $x\_y$ represents groups covering the number of objects between $(x_{i-1}, x_i]$ and the number of relations between $(y_{i-1}, y_i]$. This analysis reveals that reasoning models consistently exhibit strong performance, even as the number of relations and objects increases. Conversely, instruction-based models perform well only with limited numbers of relations and objects. In particular, when the number of objects is fixed, instruction-based models consistently perform worse as the number of relations increases.

**Analysis.** Manual inspection of the reasoning models' outputs reveals noticeable patterns. For Gemini 2 Flash Thinking, most failures are due to struggling in relation counting such as providing a lower number of relations, and producing duplicate relations, specially when the number of required relations increases. The presence of duplicate relations in the models' output can be found in Appendix Figure 6.

For other reasoning models, they tend to employ a straightforward algorithm for ordering objects, such as $A > B > C > D$, and so on. This strategy allows the models to generate an arbitrary number of consistent relations. To further test their robustness, we provide a list of objects as random strings (e.g., "dctc", "udj2", "r0c8"). Remarkably, the models maintain the simple ordering algorithm and produce consistent relations, demonstrating their robustness in maintaining logical consistency. Furthermore, we did not mention anything related to graphs in our prompt, yet approximately 88% of the DeepSeek R1 responses include the word "DAG", indicating the model understands the Plan Generation task is a graph problem. Albeit this is only one example, from an algorithm discovery perspective, the fact that models were able to figure out a straightforward solution for the generation, is an interesting indication that longer step-by-step traces can practically facilitate the discovery and execution of algorithmic approaches to problem solving.

### 4.2 CONSISTENCY DETECTION RESULTS

Next, we evaluate the Consistency Detection task. For this task, we consider 20 different groups, each defined by varying numbers of objects, relations, and the minimum depth of cycles. For example, in a list containing cycles such as $A > B > C > A$, $A > B > A$, and $B > D > A > B$, the minimum cycle depth is 2, with objects A and B involved in the shortest cycle $A > B > A$. For consistent relations, we set the minimum cycle length as 0 in the group name. For each group, we generate 20 different plans, resulting in a total of 400 samples. For each sample, the model is tasked with detecting any cycles/contradictions and, if present, outputting all of them. More details on the data groups and the prompt template can be found in Appendix.

**Overall Results.** Average F1 scores are shown in Figure 2a. Unlike the Plan Generation task, this task is more challenging for both reasoning and instruction-based models. The performance gap between the O3-mini and O1 models and the next best model (DeepSeek R1) is significant, almost 2x. Interestingly, in this task, GPT-4.5 achieves better performance than Gemini 2 Flash Thinking, indicating that an instruction-based model can outperform a reasoning model. However, the remaining instruction-based models perform significantly worse (by more than 3.8x) than the closest reasoning model.

Next, we report the fine-grained F1 scores for each model and configuration independently in Figure 2b. In this figure, the x-axis is labeled as $x\_y\_z$ represents groups covering the number of objects, the number of relations, and the minimum cycle length in the graphs. Note that every graph has a randomized numbers of cycles (capped at a maximum value), with the enforcement on the minimum cycle length. The average number of cycles across all samples is 4.37. Similar to the Plan Generation task, the larger the number of objects, relations and cycle length, the more challenging the task becomes for the models. This is evident when comparing settings with a minimum length of 0 (highlighted with red rectangles in the figure) to its subsequent columns.

**Analysis.** Upon inspecting model outputs, we observe that reasoning models (and GPT-4.5) often perform data cleaning and reformatting before answering—specifically, converting all <" relations to >" despite our intentional format shuffling for robust testing. Applying this cleaning step to GPT-4o raises its score from 2.8 to 12.9. Similarly, since DeepSeek R1 sometimes misinterprets relation direction mid-reasoning, the same trick boosts its score from 38.8 to 62.2.

Most failure cases are due to the errors in the transitive leaps ($A > B > C$ implies $A > C$), and the errors scale with the chain length. DeepSeek R1 tends to have more errors than other reasoning models from this aspect. Furthermore, DeepSeek R1 fails to adhere to a consistent algorithm to traverse the entire graph, it tends to jump to construct another part of the graph without completing all possibilities, resulting in an increased error rate when the graph is bigger and denser. This is unlike other successful cases such as O3-mini. where we observe that the model tries to traverse the complete graph before answering from its limited leaking reasoning tokens.

In addition to the patterns observed in the reasoning models, most instruct models favor a "complete" answer by producing an answer with multiple cycles even though the graph is consistent. This results in a high failure rate in tests with a consistent graph. We experiment with changing the "check contradictions" prompt by explicitly asking if the graph is consistent, but no significant change of behavior is monitored.

**Self Correction.** We then report the self-correcting performance of the models for the Consistency Detection task after prompting them with *Are you sure?*. To reduce the cost, we perform one run with all samples for all models. Figure 3a shows that most models improve their performance upon re-evaluation, with Gemini 2 Flash Thinking achieving a notable gain of over 10%. However, some models exhibit minimal change, such as O1 (0.02% reduction), while others experience a performance drop, like Llama 3.1 405B and Gemini 2.0 Pro, by 1.4% and 3.8%, respectively, the latter corresponding to a factor of $2.4x$.

Further investigation of the breakdown of output changes shows that most models do not exhibit a significant change in decision most of the time ($\geq 75\%$), Notably, Gemini 2 Flash Thinking shows the highest correction improvements over 19% of all samples, while Gemini 2.0 Pro demonstrates the most considerable performance decrease on 11.5% of all samples.

**Natural Language Framing.** Finally, Table 1a reports F1 score changes after translating the tests into natural language across different scenarios. Notably, DeepSeek R1's performance dropped substantially in both settings, with a greater decline in the consistency graph test group. Analysis of its reasoning traces reveals a marked reduction in tokens like "wait" and "?", suggesting less deliberation. We also observe more frequent directional errors (i.e., flipping $A > B$ to $B > A$), which persist and degrade the performance. O1 and O3-mini perform comparably in the code dependency scenario but struggle with the financial risk detection task.

## 4.3 COMPARISON QUESTION RESULTS

Finally, we evaluate the Comparison Question task. In this task, models are provided with a list of relations and a comparison statement, and are asked to determine whether the statement is *True*, *False*,

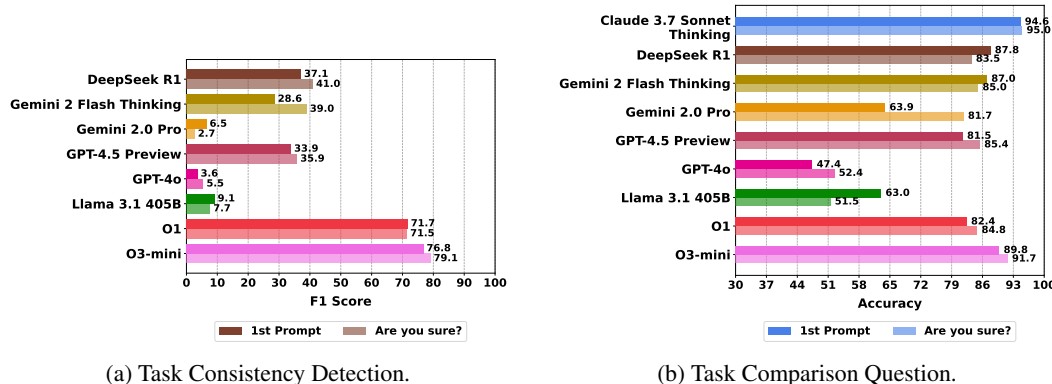

(a) Task Consistency Detection.

(b) Task Comparison Question.

Figure 3: Self correction results for the Consistency Detection and Comparison Question tasks.

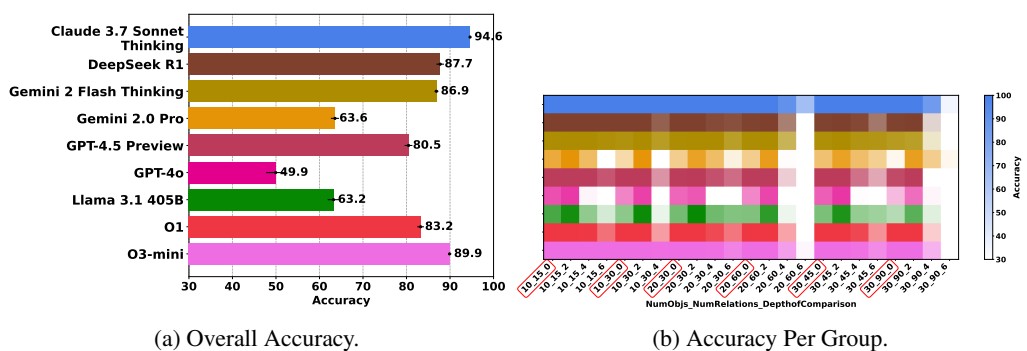

(a) Overall Accuracy.

(b) Accuracy Per Group.

Figure 4: Overview Accuracy of different models for the Comparison Question task.

or *Unknown*. For this task, we only consider consistent graphs. During data generation, we ensure diversity in graph topology, object names, and the number of $>$ and $<$ relations. We categorize the data into 23 different groups, each defined by varying numbers of objects, relations, and the minimum steps required to prove the relation in the question. In the group names, a depth of comparison of 0 indicates that the relation cannot be determined, hence the answer should be *Unknown*. This setup results in a total of 460 samples, with 20 for each group. More details on the data groups and the prompt template can be found in Appendix.

**Overall Results.** Figure 4a displays the overall accuracy. Interestingly, for this task, the O1 model drops from its second-place ranking in the previous tasks to below the Gemini 2 Flash Thinking model. Additionally, the top-performing models are much closer in performance for this task, with less than a 10% gap between the first and fifth models. Notably, GPT 4.5 outperforms other instruction-based models, and indicates a smaller performance gap between reasoning and instruction-based models in this task. However, it is important to note that the baseline for random guessing is $\frac{1}{3}$; thus, while GPT-4o achieves 49.9% accuracy, this is only approximately 16.6% better than random guessing.

Next, we present the fine-grained grouped results in Figure 4b. In this figure, the x-axis is labeled as $x\_y\_z$ represents groups covering the number of objects, the number of relations, and the depth of the comparison inference steps. Specifically we name the groups with *Unknown* comparison statement as 0 at $z$. Similar to the other tasks, as the number of objects and relations increases, the task becomes more challenging. It is also noteworthy that for instruction-based models such as Llama 3.1 405B, GPT-4o, and Gemini 2.0 Pro, it is easier to determine the correct answer when it is not *Unknown*, i.e., when the depth of comparison is greater than 0 in the chart.

**Analysis.** Analyzing the models' outputs, we found common success and failure patterns. Firstly, again, the top models rewrite mixed "$<$" and "$>$" inputs into a consistent direction before reasoning. We test GPT-4o by providing pre-aligned relations which raises its accuracy from 49.9% to 61.7%.

Table 1: Performance changes after converting the input format from symbolic ($>$ and $<$) to natural language descriptions for Consistency Detection and Comparison Question tasks.

| Model | $>$/$<$ format | Natural Language | | Model | $>$/$<$ format | Natural Language |
|---|---|---|---|---|---|---|
| DeepSeek R1 | | | | DeepSeek R1 | | |
| (Code Dependency) | **38.8** | 22.0 | | (Code Dependency) | **87.7** | 85.9 |
| (Financial Risks) | **38.8** | 20.2 | | (Financial Risks) | **87.7** | 57.4 |
| O1 | | | | O1 | | |
| (Code Dependency) | **71.3** | 74.2 | | (Code Dependency) | **83.2** | 79.8 |
| (Financial Risks) | **71.3** | 51.8 | | (Financial Risks) | **83.2** | 84.3 |
| O3-mini | | | | O3-mini | | |
| (Code Dependency) | 75.8 | **78.7** | | (Code Dependency) | **89.9** | 81.6 |
| (Financial Risks) | **75.8** | 51.3 | | (Financial Risks) | **89.9** | 85.7 |

(a) F1 score in Consistency Detection task  (b) Accuracy in Comparison Question task

A common algorithm that models tend to follow is to first list relations relevant to the two query nodes, then iteratively extend only those chains and prune irrelevant relations as it goes. Most models performe these two steps, however the main difference in result is the number of hallucinations, including incorrect chaining and picking the wrong or irrelevant evidence, both occurring more frequently with denser interconnected graphs.

Furthermore, GPT-4o frequently terminates the analysis after 2-3 hops of reasoning, concluding no direct connection exists. This results in acceptable performance at a 2-step distance but nearly complete failure at 4 and 6 steps.

**Self Correction.** To reduce the cost, we perform one run with all samples for all models. Figure 3b displays both the original results and those obtained after asking *Are you sure?*. Surprisingly, for this task, Gemini 2.0 Pro, which performed the worst in self-correction during the Consistency Detection task, shows the best improvement, with a gain of 17.8%, while Llama 3.1 405B continues to perform poorly after the self-check, with a drop of 11.5%. Other models slightly improve, except for DeepSeek R1 and Gemini 2 Flash Thinking, which show slight degradations.

Additionally, we inspect fine-grained changes for all models, and find the instruction-based models exhibit the most incorrect changes to *Unknown*, with Llama 3.1 405B and GPT-4o incorrectly converting 14.3% and 6.7% of the correct decisions to *Unknown*, respectively.

**Natural Language Framing.** Finally, Table 1b presents the changes in accuracy after we convert all tests to natural language descriptions across different scenarios. Unlike Consistency Detection, we did not observe significant performance changes from baseline except for DeepSeek R1 in financial risks dependency scenario. Similar to the Consistency Detection task, the performance degradation of DeepSeek R1 is primarily due to incorrect chaining and confusion about logical directionality when interpreting the natural language descriptions.

## 5 CONCLUSION

In conclusion, our work demonstrates that while current LLMs have made notable strides in handling structured relational reasoning, significant performance gaps remain—particularly as task complexity increases. RELEVAL not only reveals the limitations of instruction-based models in generating logically consistent plans but also highlights the robust, albeit varied performance of dedicated reasoning models. This benchmark serves as both a diagnostic tool and a catalyst for future research, guiding the development of more sophisticated architectures and self-correction strategies that can better tackle real-world, complex logical planning problems.

## REPRODUCIBILITY STATEMENT

**Implementation.** All our experimental pipelines are implemented using an open-source evaluation framework. We will make all data generation scripts publicly available upon publication, which will also enable sampling other versions RELEVAL with different problem sizes. The open source framework we use supports parameterized scripts that not only facilitate reproducibility but also allow for the generation of more challenging and customized setups tailored to specific scenarios,

Table 2: List of models studied in this study and corresponding temperature and maximum token limits used for all experiments.

| Model | temp. | max token | reasoning |
|---|---|---|---|
| Claude 3.7 Sonnet Thinking 2025-02-19 (Anthropic, 2025) | 1.0 | 32,768 | y |
| DeepSeek R1 (Guo et al., 2025) | 0.6 | 32,768 | y |
| Gemini 2.0 Pro Exp 2025-02-05 (Google, 2025a) | 1.0 | 4,096 | n |
| Gemini 2 Flash Thinking Exp 2025-01-21 (Google, 2025b) | 1.0 | 32,768 | y |
| O1 2024-12-17 (Jaech et al., 2024) | NA | 32,000 | y |
| O3-mini 2025-01-31 (high) (OpenAI, 2025) | NA | 32,000 | y |
| GPT-4o 2024-08-06 (Hurst et al., 2024) | 1.0 | 4,096 | n |
| GPT-4.5 Preview 2025-02-27 (Hurst et al., 2024) | 1.0 | 4,096 | n |
| Llama 3.1 405B (Dubey et al., 2024) | 1.0 | 4,096 | n |

such as particular object names and varying plan lengths. Additionally, the inference configurations used for all tasks are presented in Table 2, and the prompts used for each task are detailed in A.2. We experiment with different rephrasing of the prompts but found no significant effect on the outcomes. For all experiments we run, Top_p is set to 0.95, presence_penalty is set to 0 by default if the model supports this.

**Note on experimental results.** Our experiments on Gemini 2.0 Pro for Comparison Question were interrupted after four runs as the model was softly deprecated upon the release of Gemini 2.5 Pro on March 25, 2025. Tasks Plan Generation and Consistency Detection instead include results for five runs for Gemini 2.0 Pro. Additionally, due to restrictive rate limiting imposed by Claude, we were unable to complete all planned runs with Claude models before this submission. The result presented in Comparison Question is only based on one run on Claude 3.7 Sonnet with thinking model enabled.

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

# A APPENDIX

## A.1 THE USE OF LARGE LANGUAGE MODELS

We use LLMs as proof-writing assistants to enhance clarity and polish the text, identifying and correcting grammatical or stylistic issues. We also employ AI coding agents to implement the figure visualizations, with human reviewers carefully validating the results.

## A.2 DETAILED EVALUATION SETTINGS

We first present a detailed list of various object and relation numbers, together with other settings for each task.

### A.2.1 PLAN GENERATION

The combination of the number of objects and the number of relations for Plan Generation task:

```
Plan Generation data groups:
NumObjects_NumRelations

3_3,
5_5, 5_7, 5_9,
10_10, 10_15, 10_20, 10_25, 10_30, 10_35, 10_40, 10_45,
15_15, 15_24, 15_34, 15_43, 15_53, 15_62, 15_72, 15_81, 15_91
    , 15_100,
20_20, 20_29, 20_38, 20_47, 20_56, 20_65, 20_74,
20_83, 20_92, 20_101, 20_109, 20_118, 20_127, 20_136,
20_145, 20_154, 20_163, 20_172, 20_181, 20_190,
25_25, 25_34, 25_43, 25_53, 25_62, 25_71, 25_80, 25_89,
25_99, 25_108, 25_117, 25_126, 25_136, 25_145, 25_154,
25_163, 25_172, 25_182, 25_191, 25_200,
30_30, 30_39, 30_48, 30_57, 30_66, 30_75, 30_84, 30_93,
30_102, 30_111, 30_119, 30_128, 30_137, 30_146, 30_155,
30_164, 30_173, 30_182, 30_191, 30_200,
40_40, 40_49, 40_58, 40_67, 40_76, 40_85, 40_94, 40_103,
40_112, 40_121, 40_130, 40_139, 40_148, 40_157, 40_166,
40_174, 40_183, 40_192, 40_201, 40_210, 40_219, 40_228,
40_237, 40_246, 40_255, 40_264, 40_273, 40_282, 40_291, 40
    _300,
50_50, 50_59, 50_67, 50_76, 50_84, 50_93, 50_102, 50_110,
50_119, 50_128, 50_136, 50_145, 50_153, 50_162, 50_171,
50_179, 50_188, 50_197, 50_205, 50_214, 50_222, 50_231,
50_240, 50_248, 50_257, 50_266, 50_274, 50_283, 50_291, 50
    _300
```

The prompt template we use with these groups in task Plan Generation:

702
703
704
705
706
707
708
709
710
711
712
713
714
715
716
717
718
719
720
721
722
723
724
725
726
727
728
729
730
731
732
733
734
735

**Plan Generation task prompt template**

```
You are tasked with generating a list of logical and
    consistent
relationships between objects:
- Each object should have a unique name.
- The relationships use either the "<" (smaller than)
  or ">" (larger than) operator.
- The relationships must be logically consistent and acyclic
  (i.e., no cycles like A > B > A).
- Not all objects need to be directly related, but all
  relationships must be valid.

Now generate a logical and consistent list of {{num_relations
    }}
relationships with {{num_objects}} objects.

Output Format:
Return only a valid JSON object after the <RESPONSE> tag,
structured as follows:
<RESPONSE>
{
  "relationships": [
     "<object> <operator> <object>",
     "<object> <operator> <object>",
      ...
     "<object> <operator> <object>"
  ]
}
</RESPONSE>
```

736
737
738

A.2.2   CONSISTENCY DETECTION

739
740

We provide the groups of the number of objects, the number of relations, and the minium cycle length of the directed graph. We generate 20 unique graphs for each group.

741
742
743
744
745
746
747
748
749
750
751
752
753
754
755

**Consistency Detection data groups:**
**NumObjects_NumRelations_MinCycleLength**

```
10_15_0, 10_15_3, 10_15_6,
10_30_0, 10_30_3, 10_30_6,
20_30_0, 20_30_6, 20_30_12,
20_60_0, 20_60_6, 20_60_12,
30_45_0, 30_45_6, 30_45_12, 30_45_24,
30_90_0, 30_90_6, 30_90_12, 30_90_24
```

The prompt template we use in task Consistency Detection:

**Consistency Detection task prompt template**

```
Here is a list of relationships {{relationships}}, your job
    is to decide
if there are any contradictions or cycles in them, if yes (
    meaning there
are contradictions in the list), list all cycles that cause
    the
contradiction after the yes or no answer. Don't write code to
     solve
this problem. You can do whatever analysis that you need at
    the
beginning, but make sure to always finish your response with
    the
following format example, after starting with the OUTPUT tag,
     don't add
any additional text or comments.
OUTPUT:
Yes or No (Yes means there are contradictions, No means there
    is not)
1. Cycle: <A, B, C, D, E, A>
2. Cycle: <cycle>
...
N. Cycle: <cycle>
```

**Consistency Detection task self-correction prompt template**

```
Are you sure? You can do whatever analysis that you need at
    the
beginning, but make sure to always finish your response with
    the
following format example, after starting with the OUTPUT tag,
don't add any additional text or comments.
OUTPUT:
Yes or No (Yes means there are contradictions, No means there
    is not)
1. Cycle: <A, B, C, D, E, A>
2. Cycle: <cycle>
...
N. Cycle: <cycle>
```

As an example, Figure 5 illustrates an example of the graph used in the benchmark dataset. The relations represented in the prompt are $C > D, O > L, M > H, M < L, D > P, C > O, D < J, D < H, C > K, B > L, L > K, K < D, B > P, O < P, C > B$. This graph contains a single cycle with 6 nodes: $M > H > D > P > O > L > M$.

For natural language framing, here is one converted example of the software package dependency scenario:

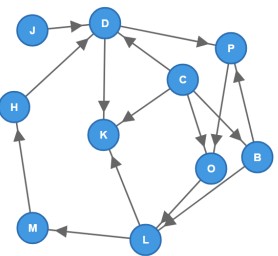

Figure 5: An example graph for the Consistency Detection task.

---

**Consistency Detection task natural language framed prompt**

```
Here is a list of relationships described in natural language
    : protoflux executes before refcast, fluxmarshal
    initializes after diskshade, nexusbloom is a prerequisite
    for protoflux, shadevault depends on refcast, nexusbloom
    is a prerequisite for diskshade, tickroll requires
    shadevault to be available first, fluxmarshal starts only
    after nexusbloom, modprobe initializes after threadmint,
    jetcache needs modprobe to be ready first, fluxmarshal
    must be loaded after tickroll, shadevault needs jetcache
    to be ready first, jetcache is needed by nexusbloom,
    diskshade initializes after jetcache, tickroll executes
    after threadmint, refcast executes before diskshade.
These relationships describe dependencies between software
    components or packages in a codebase. Your job is to
    decide if there are any contradictions or cycles in them.
     A cycle occurs when there is a circular dependency - for
     example, if package "authcore" depends on "datautil", "
    datautil" depends on "logger", and "logger" depends on "
    authcore", then there is a cycle: authcore  datautil
    logger  authcore.
In software development, circular dependencies are
    problematic because they create impossible build orders
    and can lead to compilation errors or runtime issues.
    Each component should have a clear dependency hierarchy
    without circular references.
Analyze these package dependency relationships carefully to
    identify any such circular dependencies. If yes (meaning
    there are contradictions/cycles in the list), list all
    cycles that cause the contradiction after the yes or no
    answer. Don't write code to solve this problem. You can
    do whatever analysis that you need at the beginning, but
    make sure to always finish your response with the
    following format example, after starting with the OUTPUT
    tag, don't add any additional text or comments.
OUTPUT:
Yes or No (Yes means there are contradictions, No means there
     is not)
1. Cycle: <packagename1, packagename2, packagename3,
    packagename1>
2. Cycle: <cycle>
...
N. Cycle: <cycle>
```

### A.2.3 COMPARISON QUESTION

We provide the groups of the number of objects, the number of relations, and the steps between two objects in the comparison statement.

---

**Comparison Question data groups:**
**NumObjects_NumRelations_DepthofComparison**

```
10_15_0, 10_15_2, 10_15_4, 10_15_6,
10_30_0, 10_30_2, 10_30_4,
20_30_0, 20_30_2, 20_30_4, 20_30_6,
20_60_0, 20_60_2, 20_60_4, 20_60_6,
30_45_0, 30_45_2, 30_45_4, 30_45_6,
30_90_0, 30_90_2, 30_90_4, 30_90_6
```

---

The prompt template we use in the task Comparison Question:

---

**Comparison Question task prompt template**

```
Given a set of relationship statements:
{{relationships}}

And a comparison statement to evaluate:
{{comparison}}

Your task is to determine if the comparison is True, False,
or Unknown based solely on the provided relationships.

Guidelines for evaluation:
- True: The comparison can be logically deduced from the
    given
  relationships
- False: The comparison can be proven incorrect using the
    given
  relationships
- Unknown: There is insufficient information in the
    relationships
  to definitively prove the comparison true or false

Important notes:
- Use ONLY the information provided in the relationships
- Do not make assumptions beyond what is explicitly stated
- If there are multiple possible interpretations, mark as
    Unknown
- You can do whatever analysis that you need in the response,
  but your response must end with the OUTPUT format below
- Do not add any additional text after the output

Required output format:
OUTPUT:
True/False/Unknown
```

---

**Comparison Question task self-correction prompt template**

```
Are you sure?
Note: You can do whatever analysis that you need in the
    response,
but your response must end with the OUTPUT format below.
Do not add any additional text after the output.
Required output format:
OUTPUT:
True/False/Unknown
```

For natural language framing, here is one converted example of the software package dependency scenario:

**Comparison Question task natural language framed prompt**

```
You are a software engineer analyzing code dependencies in a
    large codebase. You have been given a set of module
    dependency constraints that must be satisfied during the
    build process:
ghostbind must be loaded before indexmorph, fluxmarshal runs
    before nodemerge can execute, fluxmarshal comes earlier
    than cargowrap, indexmorph executes after logshard,
    cargowrap depends on prefx, fluxmarshal comes earlier
    than indexmorph, fluxmarshal executes before indexvisor,
    hookmesh must be ready before logshard starts, nodemerge
    comes earlier than ghostbind, hookmesh is needed by
    ghostbind, nebulock initializes after nodemerge,
    fluxmarshal initializes after logshard, nebulock executes
     before ghostbind, logshard must be loaded before prefx,
    fluxmarshal initializes after prefx.
A colleague has asked you the following question about the
    dependency order:
Do we need to load cargowrap before indexmorph?
Your task is to determine if the answer is True, False, or
    Unknown based on the dependency constraints provided
    above.
Analysis Guidelines:
- True: The dependency relationship can be definitively
    proven from the given constraints (either directly stated
     or through transitive dependencies)
- False: The dependency relationship contradicts the given
    constraints or would create a circular dependency
- Unknown: The given constraints do not provide sufficient
    information to determine the dependency relationship
Important considerations:
- Dependencies are transitive: if A depends on B and B
    depends on C, then A indirectly depends on C
- Circular dependencies are not allowed in the build system
- Only use the explicitly stated dependency constraints - do
    not make assumptions about unstated relationships
- Consider all possible dependency paths when determining
    transitivity
- If the constraints lead to ambiguous or conflicting
    dependency chains, answer Unknown
Please analyze the dependency relationships step by step,
    then provide your final answer in the required format
    below.
Required output format:
OUTPUT:
True/False/Unknown
```

## A.3 DETAILED TASK RESULTS

### A.3.1 DETAILED PLAN GENERATION TASK RESULTS

Figure 9, Figure 10, Figure 11 and Figure 12 show the detailed result of Plan Generation task. The rows are the number of objects, and the columns are the number of relations prompted to the model. The empty cells are the combinations that are not covered in this test. Each colored cell shows the average accuracy result of 3 runs.

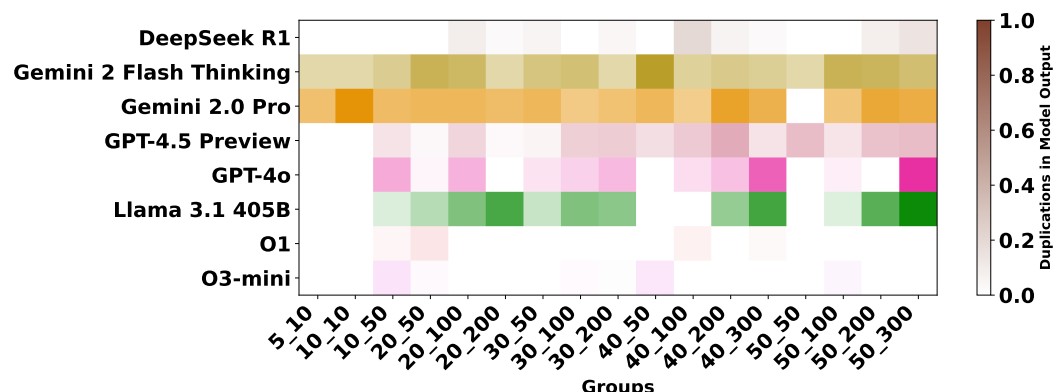

Figure 6: Plan Generation Relation Duplications in Model Output

We also notice that some models are easier to output duplicated relations in general, like Gemini models. And some models get easier to output duplicate relations as the task gets harder, such as GPT-4o and Llama 3.1 405B. Figure 6 shows the detailed results of the average existence of duplications (True of False) in model's output for each model and task groups.

### A.3.2   DETAILED CONSISTENCY DETECTION TASK RESULTS

Figure 7 shows the detailed result for Consistency Detection task. The average F1 score is aggregated based on the group of the number of objects, the number of relations and the minimum cycle length in the provided relations to the model. We performed 5 runs for each model, and calculated the average F1 score.

### A.3.3   DETAILED COMPARISON QUESTION TASK RESULTS

Figure 8 shows the detailed result for Comparison Question task. The average accuracy is aggregated based on the group of the number of objects, the number of relations and the depth of the provided comparison statement to the model. Except for Claude 3.7 Sonnet, We performed 5 runs for each model, and calculated the average.

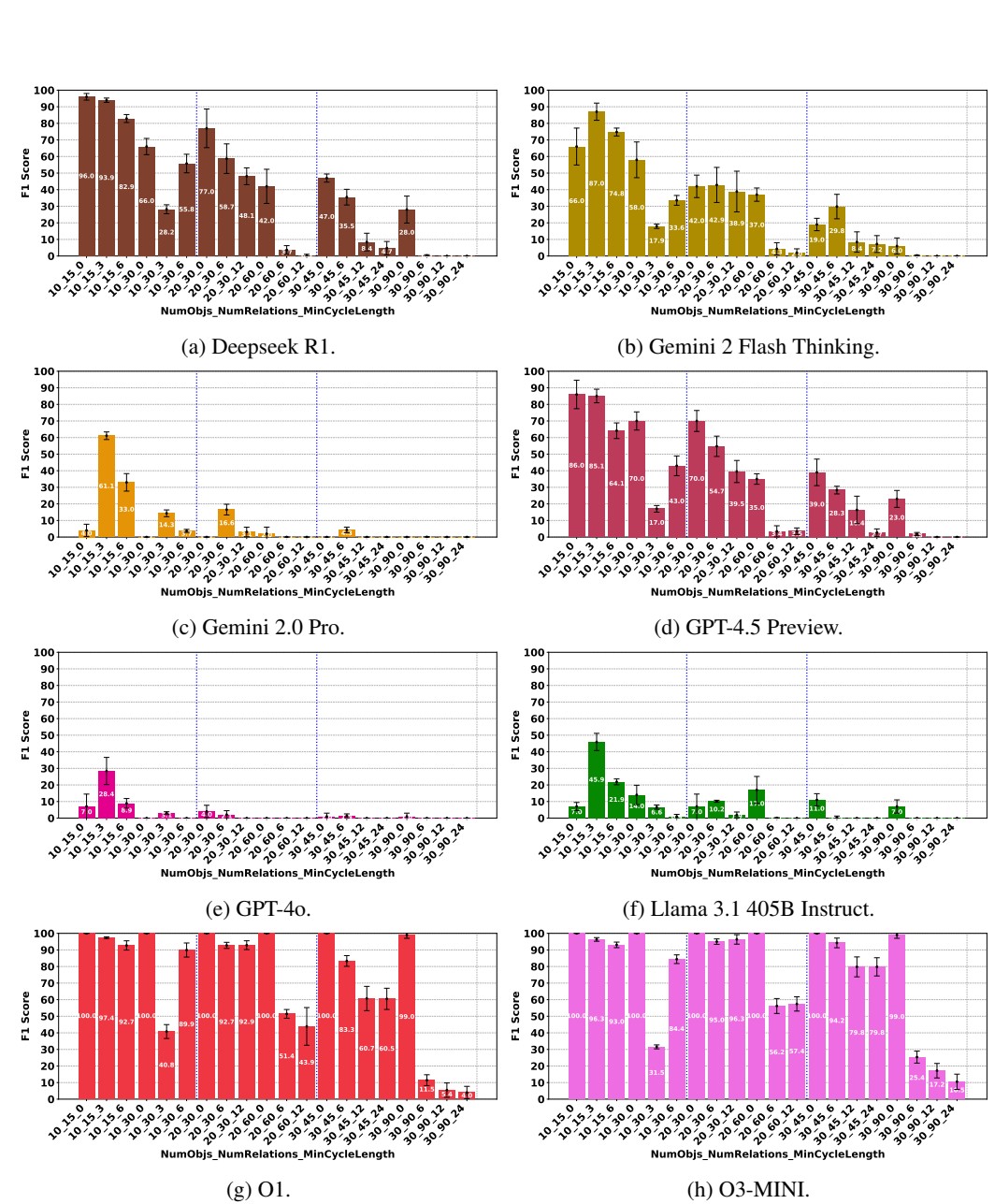

Figure 7: Detailed Average F1 Score of Consistency Detection Task - Continued

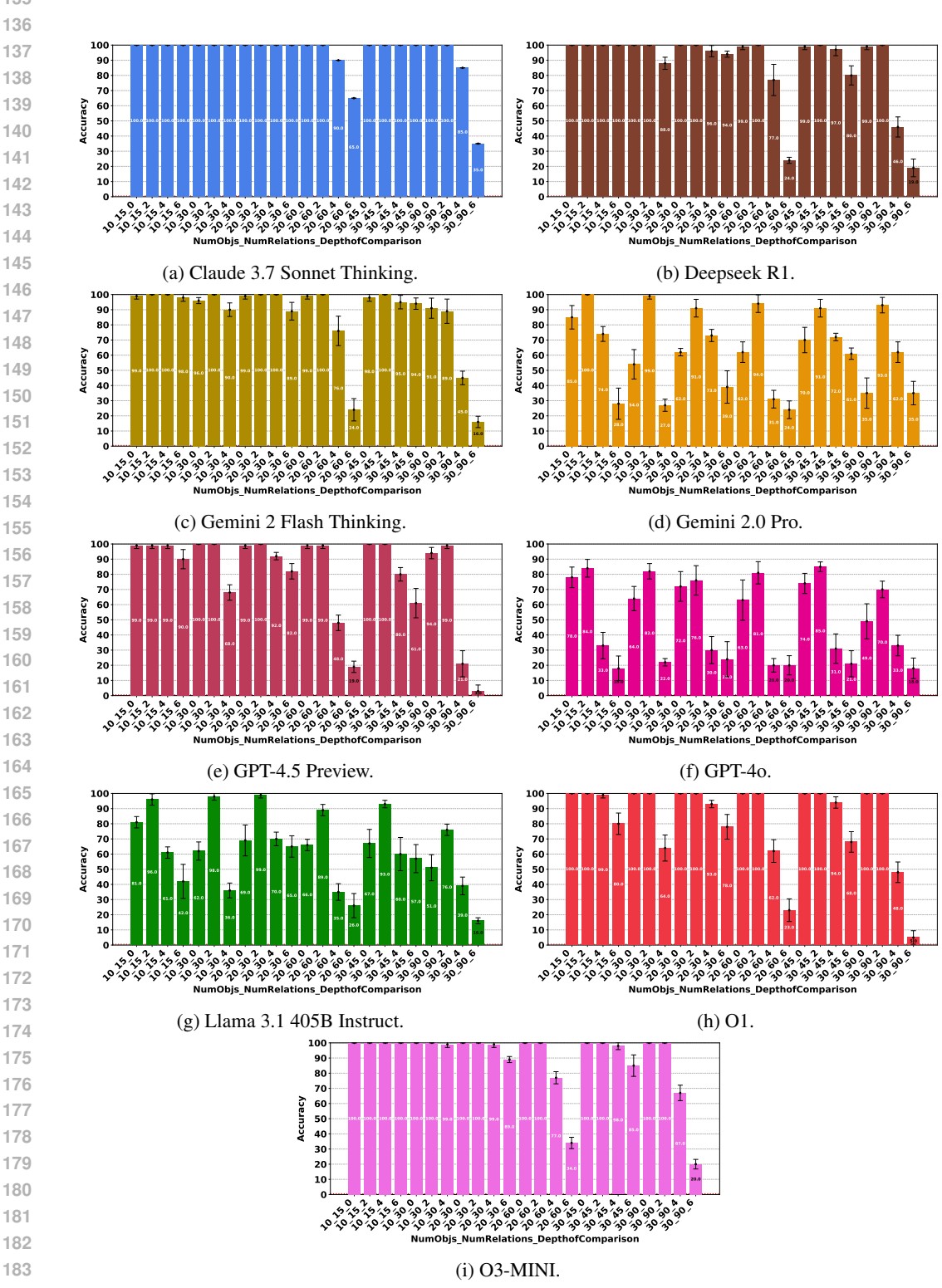

(a) Claude 3.7 Sonnet Thinking.

(b) Deepseek R1.

(c) Gemini 2 Flash Thinking.

(d) Gemini 2.0 Pro.

(e) GPT-4.5 Preview.

(f) GPT-4o.

(g) Llama 3.1 405B Instruct.

(h) O1.

(i) O3-MINI.

Figure 8: Detailed Average Accuracy of Comparison Question Task

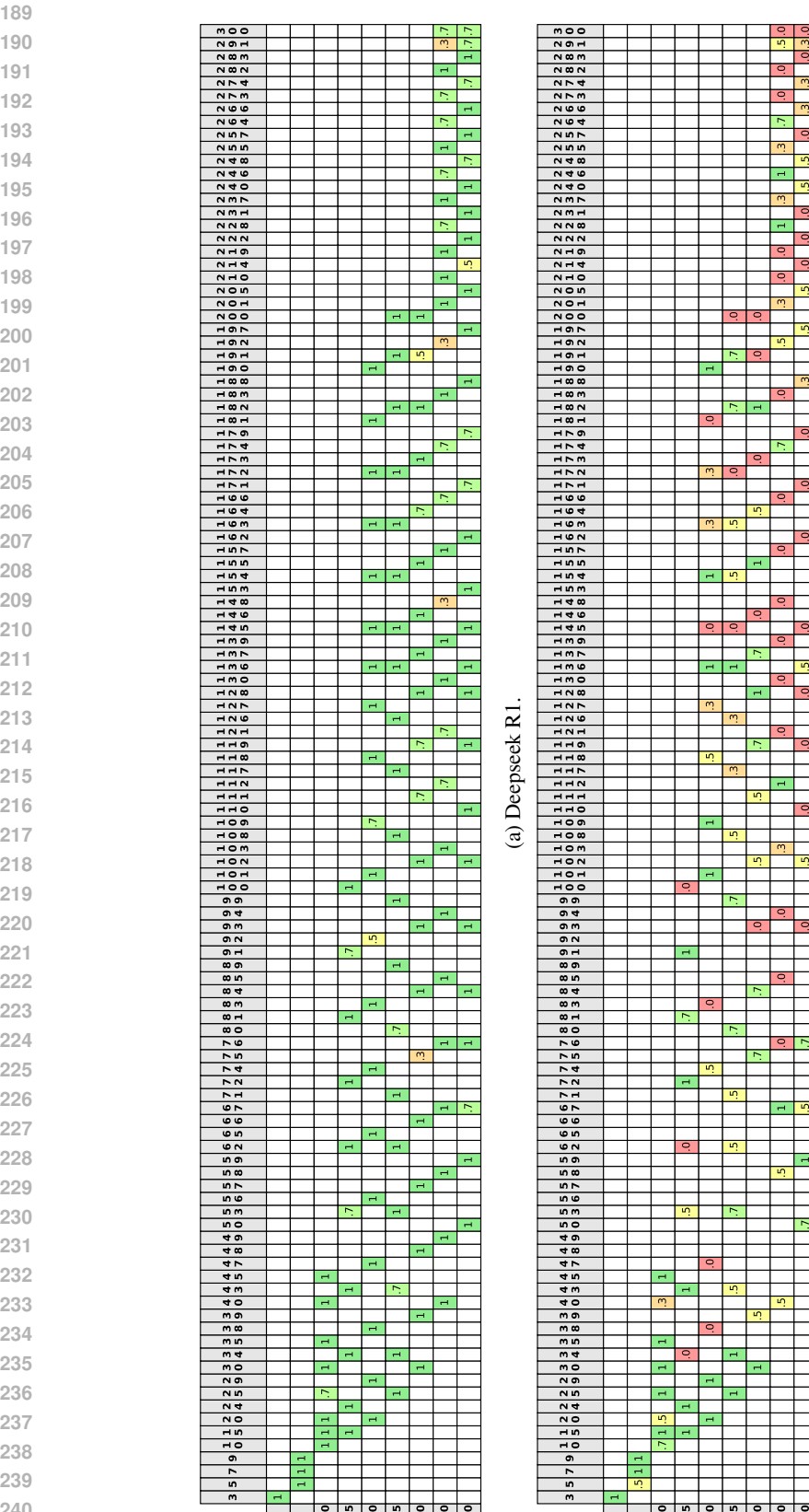

(a) Deepseek R1.

(b) Gemini 2 Flash Thinking.

Figure 9: Detailed average accuracy of Plan Generation task

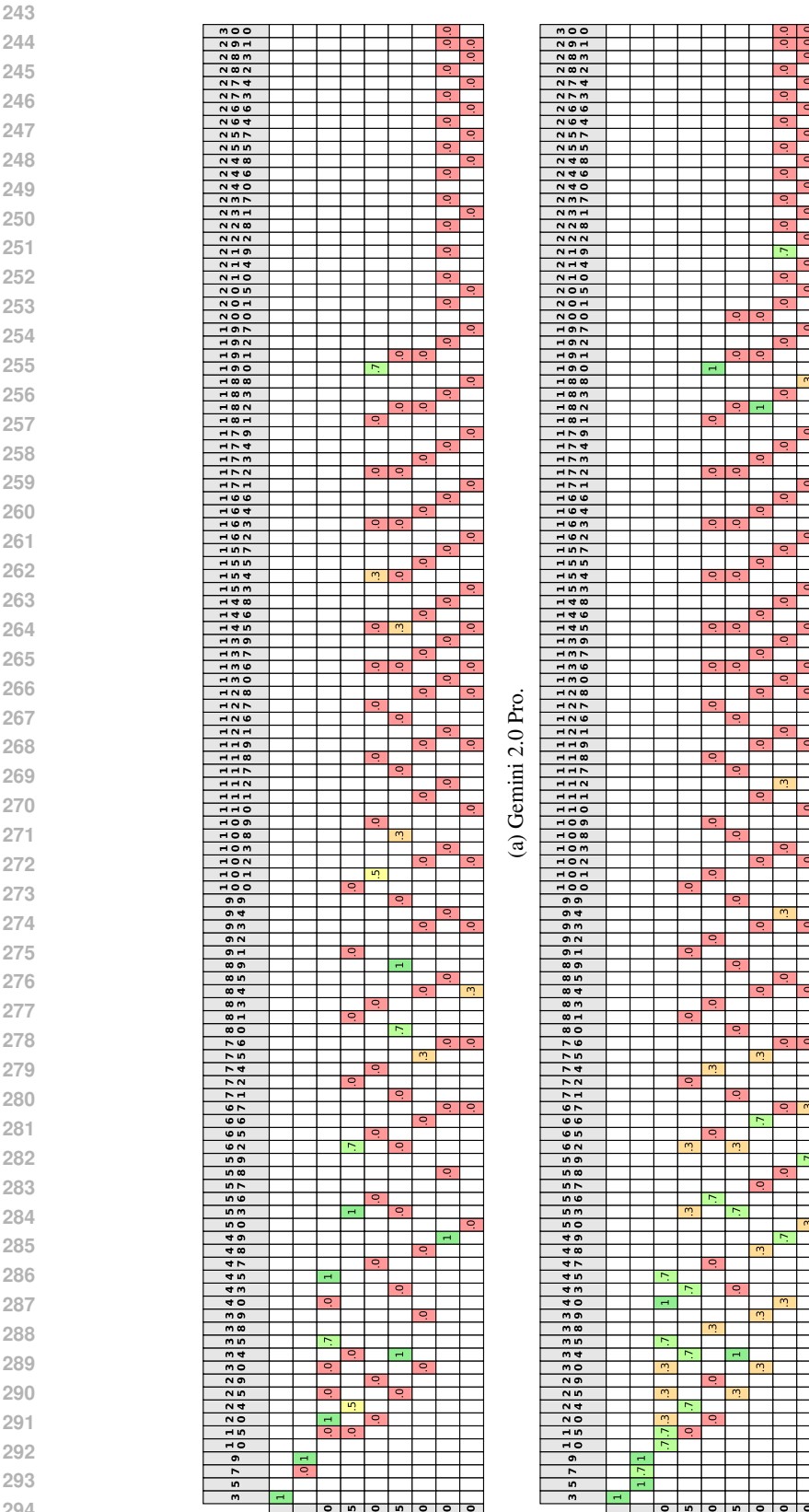

Figure 10: Detailed average accuracy of Plan Generation task - Continued

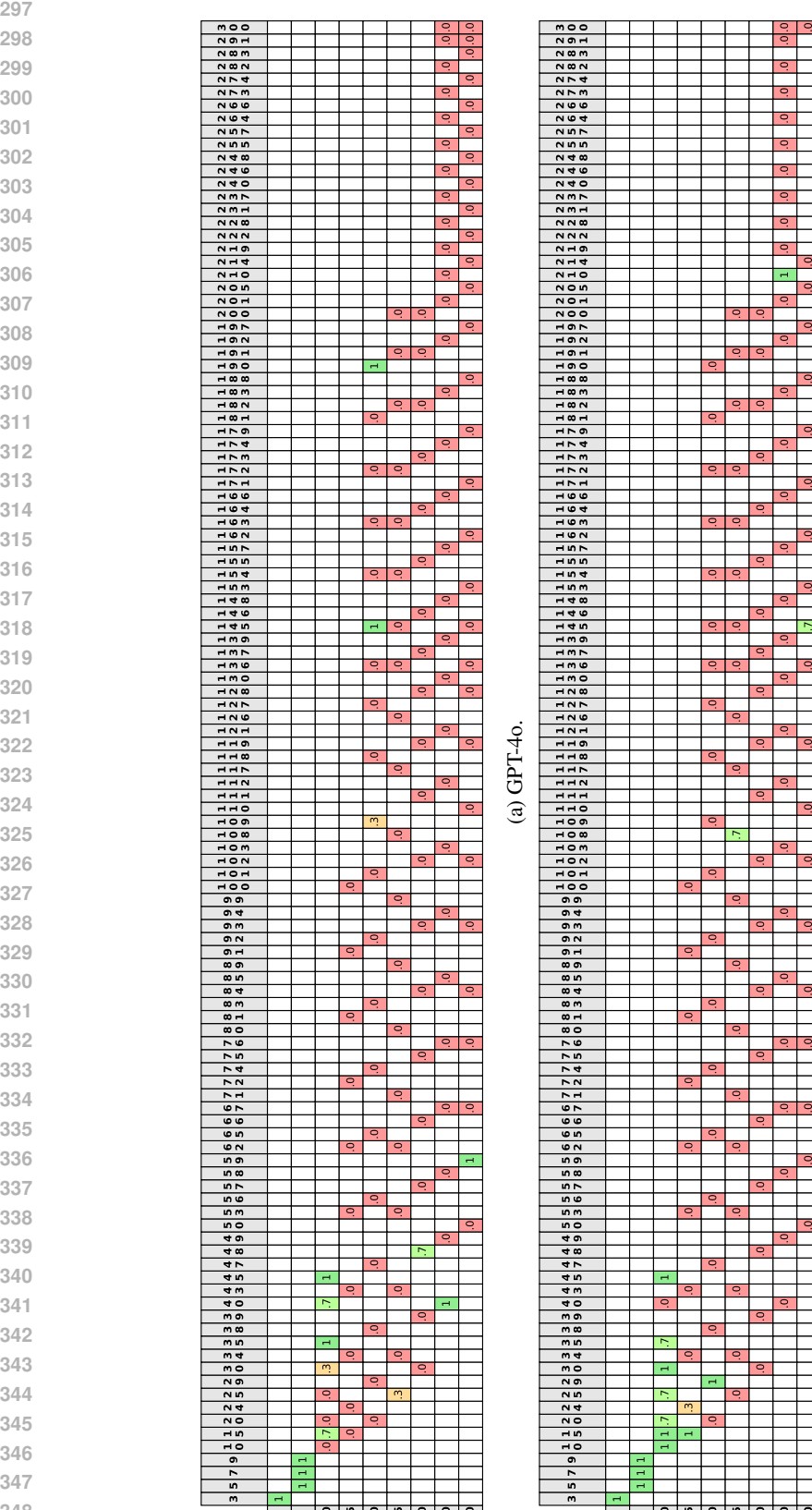

(a) GPT-4o.

(b) Llama 3.1 405B Instruct.

Figure 11: Detailed average accuracy of Plan Generation task - Continued

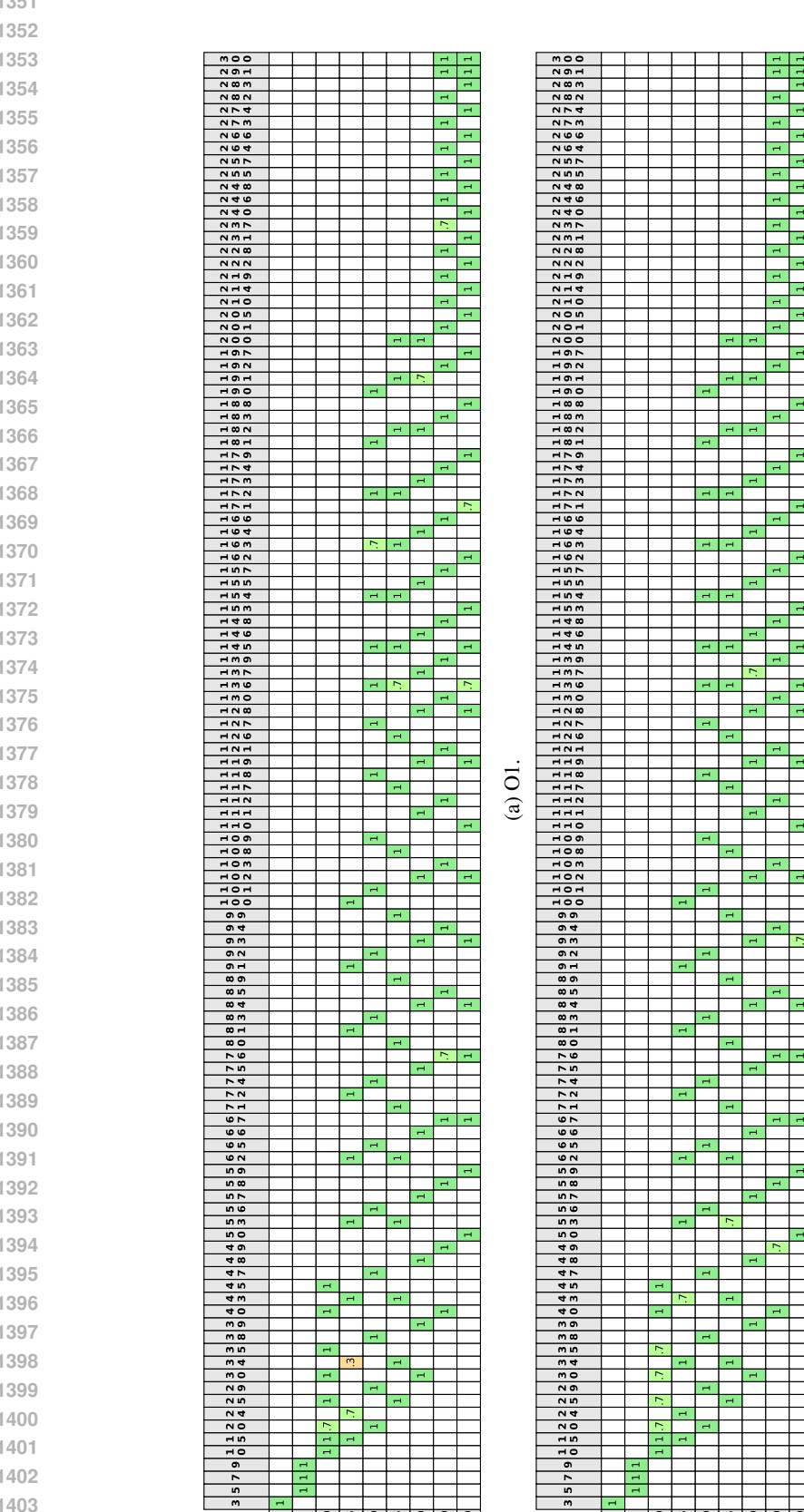

(a) O1.

(b) O3-MINI.

Figure 12: Detailed Average Accuracy of Plan Generation Task - Continued

