# OpenReview forum: "RelEval: A Structured Benchmark for Logical and Relational Reasoning in LLMs"
_ICLR.cc/2026/Conference — ICLR 2026 Conference Withdrawn Submission_

### Official Review · Reviewer_Vo9c · 2025-10-26

**Soundness:** 3
**Presentation:** 3
**Contribution:** 2
**Rating:** 4
**Confidence:** 3

**Summary:**

The paper proposes a mechanically generated benchmark for Large Language Models, and evaluates several state-of-the-art LLMs against it.

**Strengths:**

The benchmark construction technique of the paper may have some merit in itself. However, it is overall very similar to previous work in my opinion (more on this in "Weakness").

**Weaknesses:**

The idea of mechanically generating logical reasoning benchmarks has been explored extensively in the literature. The paper should thus include at least some comparison between the proposed benchmark and existing ones. Especially, task (3) comparison questions seems very similar to such benchmarks as HotpotQA. It is unclear to me if task (1) and (2) differ fundamentally from existing multi-hop QA benchmarks. I am happy to reassess the novelty of these tasks if the authors can provide more prompts and model outputs.

**Questions:**

Please see “Weaknesses”.

---

### Official Review · Reviewer_VRje · 2025-10-31

**Soundness:** 2
**Presentation:** 2
**Contribution:** 2
**Rating:** 2
**Confidence:** 4

**Summary:**

The paper proposes RelEval, a benchmark for evaluating logical and relational reasoning in LLMs. The benchmark is based on dynamically generated relational graphs of controllable difficulty, varying the number of objects and relations. It comprises three tasks: (1) Plan Generation, where models must construct a valid, acyclic graph given object and relation counts; (2) Consistency Detection, where models must identify cycles or contradictions in a given graph ; and (3) Comparison Question, where models must validate a specific relation as True, False, or Unknown.
The authors evaluate differnet LLMs, categorizing them as "instruction-based" or "reasoning-focused". The results show that the reasoning models generally outperform the instruction models, but all models struggle as the problem complexity (e.g., number of objects, chain depth) increases.

**Strengths:**

- The paper focuses on an important problem of evaluating relational reasoning.
- The organization of the paper structure is clear.

**Weaknesses:**

- Lack of Novelty: The benchmark tasks seems to be simply re-formulations of classic, well-understood computer science problems. "Consistency Detection" is graph cycle detection. "Comparison Question" is transitive-closure-based path queries. "Plan Generation" is simple DAG generation. The related work section itself cites many benchmarks that already test multi-hop reasoning and planning. The paper does not adequately differentiate its contribution from existing benchmarks for multi-hop reasoning and planning.

- Lack of Significance: The paper's main finding is that LLMs are not good at solving algorithmic graph problems. This is not a surprising or insightful result; it is the expected behavior for models that are not designed to be algorithmic graph processors.

**Questions:**

- Regarding significance, could the authors provide a compelling argument for why a transformer should be able to perform, for instance, cycle detection on 30 nodes and 90 edges?

- Could the authors discuss related works in the field relational deep learning and make a comparision?

- Could the authors provide a more rigorous standard regarding the split between "reasoning" and "instruction" models in their experiments?

---

### Official Review · Reviewer_g6p3 · 2025-11-01

**Soundness:** 2
**Presentation:** 1
**Contribution:** 2
**Rating:** 2
**Confidence:** 4

**Summary:**

This paper introduces RelEval, a benchmark designed to evaluate LLMs on logical and relational reasoning. The benchmark comprises three tasks: plan generation, consistency detection, and comparison question. The study evaluates a broad range of models, including both instruction-based and reasoning models. It further explores models’ self-correction behavior and extends the evaluation to natural language framing of these tasks. Empirical results reveal substantial performance gaps between reasoning and instruction-based models. The paper is also supported by qualitative inspection of reasoning traces that highlight specific model failure modes.

**Strengths:**

- The benchmark covers three related tasks that represent three dimensions of logical and relational reasoning: generation, checking, and question answering.
- The experiments include a diverse set of models, covering both instruction-based and reasoning ones.
- The manual inspection of reasoning traces is quite extensive and provides useful insights for practitioners into how models fail and reason differently.

**Weaknesses:**

- The benchmark tasks are limited in capturing the broad range of logical and structural reasoning. Although the three tasks aim to test different aspects, they still center on one specific type of reasoning: constructing and analyzing relational graphs, and the logic is mainly expressed through simple “<” and “>” relations. This narrow focus restricts the scope of what the benchmark truly measures compared to its broader claim of evaluating logical and relational reasoning, as the title and the introduction suggests.
- The position and contribution of the benchmark in existing work is unclear. The topic of this benchmark appears to encompass both logical reasoning and relational/graph reasoning. However, given the abundance of existing benchmarks addressing these areas and covering more comprehensive logical types and graph problems, the contribution of this work seems limited.
- The verification process is not discussed clearly. According to the appendix, the benchmark uses JSON output for evaluation. It is not explained whether the evaluation is fully automated or how format errors are handled. Since models often fail to follow strict JSON formatting, this could affect accuracy, and more discussion or analysis of this impact would be useful.
- Although natural language framing is included, it is still translated from the synthetic tasks. Given the paper’s motivation in real-world reasoning domains like supply chain, using more realistic, domain-specific examples would make the evaluation stronger and more convincing.
- The empirical results could be more meaningful if they were organized into clearer conclusions, rather than buried within the overall results and analysis. The overall writing of the paper needs to be improved for better readability and clarity.

**Questions:**

- Besides increasing the number of objects and relations, is there a way to ensure greater diversity in the reasoning traces being evaluated?
- When generating relational graphs, what types of graph structures are used?

---

### Official Review · Reviewer_VFyz · 2025-11-01

**Soundness:** 2
**Presentation:** 2
**Contribution:** 2
**Rating:** 2
**Confidence:** 2

**Summary:**

The paper introduces RELEVAL, a benchmark for logical and relational reasoning of LLMs, and includes three graph tasks: (1) Plan Generation: generate a consistent set of directed comparative relations under constraints), (2) Consistency Detection: detect and enumerate cycles, and (3) Comparison Question: judge a relation as True/False/Unknown given a graph. It also probes self-correction using an “Are you sure?” follow-up prompt and provides a natural-language framing of existing tasks. Experiments cover instruction-tuned and reasoning-oriented models and showed that Consistency Detection is challenging for LLMs.

**Strengths:**

- The paper clearly articulates each task, the evaluation metrics, and the data-generation pipeline, making the setup easy to follow.
- For consistency detection, the choice to mix notations and shuffle statement order is well-motivated and helps discourage shortcut pattern matching. This is a thoughtful input design.

**Weaknesses:**

- I am mainly concerned about the novelty and position of this dataset. Several proposed tasks appear to map classic graph problems. For example, the plan-generation task resembles DAG construction, and the consistency-detection task aligns with cycle detection (with “minimum cycle length” achieved by cycle enumeration). Given the current focus on three question types, it’s hard to assess whether this is truly “the first benchmark” that jointly evaluates graph generation and downstream reasoning, especially since prior work covers related capabilities.
- The qualitative results are informative but largely model- and task-specific (e.g., “GPT-4o terminates after 2–3 hops”), which limits generalization. This paper would benefit from any error mode or causal factors analysis that generalizes beyond the presented settings.
- The NL variant currently feels close to the formal framing. The example on p.16 mainly relabels nodes as letter strings and adds brief context, which may not fully exercise NL understanding as it doesn't use any linguistic features (such as ambiguity in NL vs. formal description).

**Questions:**

- It would be helpful to clarify what capability is uniquely measured here that prior work could not measure with tweaks and adjustments.
- The self-correction mechanism seems to reduce performance for some models but not others. I am curious what some hypotheses are about this phenomenon.

---

### Note · Authors · 2025-11-27

I have read and agree with the venue's withdrawal policy on behalf of myself and my co-authors.